# Impacts of Extreme Climate Events on Future Rice Yields in Global Major Rice-Producing Regions

**DOI:** 10.3390/ijerph19084437

**Published:** 2022-04-07

**Authors:** Weixing Zhao, Jieming Chou, Jiangnan Li, Yuan Xu, Yuanmeng Li, Yidan Hao

**Affiliations:** 1State Key Laboratory of Earth Surface Processes and Resource Ecology, Faculty of Geographical Science, Beijing Normal University, Beijing 100875, China; 201921051146@mail.bnu.edu.cn (W.Z.); 201921051150@mail.bnu.edu.cn (J.L.); 202031051081@mail.bnu.edu.cn (Y.X.); 202021051153@mail.bnu.edu.cn (Y.L.); 202121051159@mail.bnu.edu.cn (Y.H.); 2Southern Marine Science and Engineering Guangdong Laboratory (Zhuhai), Zhuhai 510275, China

**Keywords:** extreme climate events, rice yields, rice-producing regions, food security risk

## Abstract

Under the dual impacts of climate change and COVID-19, there are great risks to the world’s food security. Rice is one of the three major food crops of the world. Assessing the impact of climate change on future rice production is very important for ensuring global food security. This article divides the world’s main rice-producing regions into four regions and uses a multivariate nonlinear model based on historical economic and climatic data to explore the impacts of historical extreme climatic events and economic factors on rice yield. Based on these historical models, future climatic data, and economic data under different shared socioeconomic pathways (SSPs), the yields of four major rice-producing regions of the world under different climate change scenarios (SSP126, SSP245, and SSP585) are predicted. The research results reveal that under different climate change scenarios, extreme high-temperature events (Tx90p) and extreme precipitation events (Rx5day, R99pTOT) in the four major rice-producing regions have an upward trend in the future. Extreme low-temperature events (Tn10p) have a downward trend. In the rice-producing regions of Southeast Asia and South America, extreme precipitation events will increase significantly in the future. The prediction results of this model indicate that the rice output of these four major rice-producing regions will show an upward trend in the future. Although extreme precipitation events will have a negative impact on rice production, future increases in rice planting areas, economic development, and population growth will all contribute to an increase in rice production. The increase in food demand caused by population growth also brings uncertainty to global food security. This research is helpful for further understanding climate change trends and risks to global rice-production areas in the future and provides an important reference for global rice-production planning and risk management.

## 1. Introduction

According to the 2020 report of the Food and Agriculture Organization (FAO), nearly 750 million people worldwide faced severe food insecurity in 2019, accounting for nearly one-tenth of the world’s total population [1]. Guaranteeing food security is an important foundation for the sustainable development of the world’s economy and human society. Rice is one of the three most important food crops in the world and is widely distributed throughout the world. Rice is the main food source for more than 50% of the world’s population and is grown in 122 countries [2,3]. Future changes in rice production will have a significant impact on the global grain production pattern and food security [4].

Under the dual impacts of climate change and the new coronavirus epidemic, there are great risks to global food security. Studies indicate that the new coronavirus (COVID-19) pandemic caused the number of undernourished people in the world to increase from 83 million to 132 million in 2020 [1]. There is greater uncertainty about the impact of climate change on future rice yields. Although climate change can prolong the growth period of crops, it can also shorten the growth period and reduce the growth amount, which reduces the potential of crop production and affects crop yield [5,6,7]. The increase and enhancement of extreme climate events caused by climate change is the main threat to rice yield in the future. The Special Report on Climate Change and Land Use by the Intergovernmental Panel on Climate Change (IPCC) pointed out that the global average annual cereal yield loss is approximately 10% due to extreme weather events [8].

Under the background of global climate change, the frequency, intensity, and duration of extremely high temperatures, extreme precipitation, and drought all show a significant increase at the global scale, becoming some of the most severe challenges facing food production [9,10,11,12,13,14,15]. In recent years, studies based on historical data analysis have indicated large interannual fluctuations in the production of the world’s three major staple foods from 1979 to 2008, of which 32% to 39% were caused by climate change, while extreme climate events occurred in South Asia and China and determined grain yield fluctuations in much of the United States [16]. Furthermore, some studies have emphasized the important impacts of extreme climate events on rice yield. For example, in Jiangsu Province and Hainan Province, China, extreme precipitation indicators were significantly negatively correlated with rice yield [11,17,18], but these studies only considered extreme precipitation. Rice yield is affected by multiple extreme climatic events, such as extremely high temperatures, droughts, extremely low temperatures, and extreme precipitation, during the growing period [19]. How to scientifically assess the comprehensive impact of multiple extreme climate events on global rice yield, especially the degree of impact, mechanisms, and spatial patterns of extreme climate events on global rice yield, is an urgent problem to be solved.

In addition to assessing historical food production with respect to climate change, predicting the impacts of climate change on future food production has always been a focus of the scientific community. Many studies have used simulation data under different shared socioeconomic pathways (SSPs) of the Coupled Model Intercomparison Project Phase 6 (CMIP6) combined with crop models to evaluate the impacts of future climate change on the yield of major food crops [20,21]. However, most studies are based on a small area, and most crop models are mainly based on changes in average climatic states, making it difficult to simulate the sensitivity of crop yields to extreme climate events. This leads to large deviations in crop yield forecasts, hindering scientific assessments of the impacts of extreme climate change on global food security. Considering the importance of extreme climatic events to rice production, an in-depth analysis of the comprehensive impact of extreme climatic events on future rice yields and the importance of future agricultural forecasting and adaptation planning is needed.

Climate change has a significant impact on rice yield, but future changes in economic factors cannot be ignored. In the future, economic factors, such as the area of grain sown, the total power of agricultural machinery, and agricultural labor, will have a significant impact on rice production [21,22,23,24]. How to combine economic and climatic factors to comprehensively evaluate the future rice yield is a scientific problem that urgently needs to be solved.

Based on historical economic data and historical climate data, this paper uses statistical yield models to explore the extent to which rice yields in the world’s four major rice-producing regions were affected by historical extreme climate events and to model the impacts of extreme climate events and economic factors on the rice yields. Using CMIP6 climate data, future changes in extreme climate events in the four major rice-producing regions in the world were predicted. Based on the historical model, the future economic data under the shared socioeconomic pathway 2 (SSP2) and the extreme climate index under different climate change scenarios were selected to evaluate the future rice yield of the four major rice-producing regions in the world. This will help to further understand climate change trends and risks in rice-producing regions globally and has important scientific and practical value for global rice-production planning and risk management.

## 2. Materials and Methods

### 2.1. Reanalysis Data and CMIP6 Data

The historical reanalysis data were from ERA5. ERA5 is the latest generation of reanalysis data created by the European Centre for Medium-Range Weather Forecasts (ECMWF). In comparison to previous versions, ERA5 has a higher spatial and temporal resolution, with a temporal resolution of 1 hour and a spatial resolution of 0.25° × 0.25°. In addition, the data cover the historical period from 1950 to the present.

The future climate scenario data were obtained from the climate model data of the Coupled Model Intercomparison Project Phase 6 (CMIP6) (https://esgf-node.llnl.gov/projects/cmip6/, accessed on 21 May 2021). The data used in this study were daily near-surface air temperature, daily maximum near-surface air temperature, daily minimum near-surface air temperature, and daily precipitation. The selected scenarios were SSP126, SSP245, and SSP585. The data selection time period was 1950–2100. Fundamental information on the selected models is listed in Appendix A.

### 2.2. Historical and Future Economic Data

#### 2.2.1. Rice Yield Data

The Global Historical Yield Dataset (GDHYv1.2 + v1.3) provides annual data on 0.5-degree grid cell yield estimates for major global crops for the period 1981–2016 [25]. The crops considered in this dataset are corn, rice, wheat, and soybean. This study mainly selected the annual yield data for rice.

#### 2.2.2. Population, GDP, and Land-Use Data

Population, gross domestic product (GDP), and land-use data were primarily derived from historical and future economic data from the Inter-Sectoral Impact Model Intercomparison Project (ISIMIP). The project collects a large amount of forecast data on the future economic climate.

Population and gross domestic product (GDP) data were selected from the future shared socioeconomic pathways (SSPs) developed by Daisuke Murakami et al. [26] and Geiger, T. et al. [27]. These data take into account the spatial and socioeconomic interactions between cities and are widely used in future socioeconomic research.

The future land-use data were selected from the arable land area dataset of 15 future grain crops developed by Popp et al. [28,29]. This dataset is based on the MAgPIE land-use model for future land-use predictions based on the SSP2. The spatial resolution of these data is 0.5° × 0.5°. This dataset contains the past and future percentage of arable land area for 15 basic crops in the grid points. The sum of the non-irrigated rice planting area and the irrigated rice planting area was selected as the rice planting area in this study.

### 2.3. Study Area and Selection of Indicators

The global rice-producing region was selected as the study area for this paper. According to the United Nations Food and Agriculture Organization (FAO) statistical report, the world’s rice-producing regions are mainly concentrated in Asia, with nearly 90% of the world’s rice sowing area and 91% of the world’s rice output. South America (3.2%), Africa (2.9%), and North America (1.4%) represent additional rice sowing areas, and Central America, Europe, and Oceania together account for only 2.5% of the world’s rice sowing area, mainly in Australia. Combined with the sown area of rice, the world is divided into four major rice-producing regions, namely, the (I) Southeast Asia rice-producing region (SEA), (II) African rice-producing region (AF), (III) South American rice-producing region (SA), and (IV) North American rice-producing region (NA) (Figure 1).

The selection of indicators of extreme climate events was mainly based on 27 typical climate indices defined by the Expert Team on Climate Change Detection and Indices (ETCCDI), jointly established by the World Meteorological Organization (WMO) and other organizations. The number of warm days (number of days where the daily maximum temperature was higher than the 90% quantile value Tx90p, %), number of cold night days (number of days where the daily minimum temperature was lower than the 10% quantile value Tn10p, %), total amount of heavy precipitation (annual cumulative precipitation with daily precipitation greater than the 99th percentile, R99p, mm), and the 5-day maximum precipitation (maximum precipitation for 5 consecutive days, Rx5day, mm) were selected. The future socioeconomic paths SSP126, SSP245, and SSP585 were selected. In this study, we used 1981–2015 as the historical reference period, 2040–2060 as the mid-21st century period (denoted 2050), and 2080–2100 as the end of the 21st century period (denoted 2100).

### 2.4. Research Methods

#### 2.4.1. Statistical Model

To assess the contribution of economic factors and extreme climate events to rice yields in various global production regions, we first normalized the data. We used the highest value normalization (Normalization) to map all data between 0 and 1. The specific method was as follows:(1)$$X=\frac{X_{i}-\min\left(X\right)}{\max\left(X\right)-\min\left(X\right)}$$

Then, to analyze the contribution of extreme climate events to the rice yield in the top four rice-producing regions of the world, we used the following statistical yield models [30,31,32]:(2)$${\mathrm{Yield}}_{yi}=\sum_{j=1}^{s}\left(\alpha_{j1}+\alpha_{j2}\times X_{yij}+\alpha_{j3}\times X_{yij}^{2}\right)$$
where *y* is different rice-producing regions, *i* is different years, *j* is different variables selected (including the selected extreme climate index and economic index), *s* is the number of selected indicators, and $X_{yij}$ is the value for the rice-producing region *y* in the year *i* with the variable index *j*.

#### 2.4.2. Flowchart

Figure 2 is the flow chart of this study. We use a multivariate nonlinear model based on historical economic and climatic data (ERA5 Historical Data) to explore the impacts of historical extreme climatic events and economic factors on rice yield. Based on these historical relationship models, future climatic data, and economic data under different shared socioeconomic pathways (SSPs), the yields and climate change risk of four major rice-producing regions under different climate change scenarios (SSP126, SSP245, and SSP585) were predicted.

## 3. Results

### 3.1. Comparison of CMIP6 Data and Reanalysis Data

From the perspective of regional distribution, under the moderate-emissions shared socioeconomic pathway (SSP245) (Figure 3), the number of warm days (Tx90p), total amount of extremely heavy precipitation (R99p), and maximum precipitation in 5 days (Rx5day) were significantly higher than those in the historical reference period and showed increasing trends in the future. The number of cold nights (Tn10p) was lower than that in the historical reference period, showing a decreasing trend.

The number of warm days (Tx90p) was significantly higher in the South American rice-producing region (SA) and African rice-producing region (AF) than in the other rice-producing regions, and the number of future warm days (Tx90p) was significantly increased in almost all the rice-producing regions. The number of cold nights (Tn10p) decreased significantly in the South American rice area (SA), the western North American rice area (NA), the eastern Southeast Asian rice area (SEA), and the African rice area (AF). This was almost in the same areas as those where the number of warm days (Tx90p) increased.

The total amount of extremely heavy precipitation (R99p) projected for the future did not change significantly in the four rice-producing regions, showing a slightly increasing trend. The total future extreme precipitation (R99p) increased most significantly in the northeastern part of the South American rice-producing region (SA). The future distribution of the maximum precipitation on the fifth day (Rx5day) in the four rice-producing regions was similar to the distribution of the total heavy rainfall (R99p). Extreme heavy precipitation was projected to increase in the northeastern part of the South American rice-producing region (SA) in the future.

Appendix A show changes in extreme climate events under the low-emissions shared socioeconomic pathway (SSP126) and high-emissions shared socioeconomic pathway (SSP585), and the number of warm days in the future rice-production areas under the low-emissions pathway (Tx90p), the total amount of extremely heavy rainfall (R99p), and the maximum rainfall on the fifth day (Rx5day) increased slowly. Under the high-emissions pathway, the number of warm days (Tx90p), total amount of extremely heavy precipitation (R99p), and 5-day maximum precipitation (Rx5day) in the rice-producing region increased significantly in the future. There was an opposite change trend for the number of cold nights (Tn10p).

From the regional averages of the four major rice-producing regions, under the moderate-emissions shared socioeconomic pathway (SSP245) (Figure 4), the number of warm days (Tx90p), total amount of extremely heavy precipitation (R99p), and 5-day maximum precipitation (Rx5day) were projected to increase in the future. The number of cold nights (Tn10p) showed a decreasing trend in the future.

The number of warm days in the future (Tx90p) was significantly higher in the South American rice-producing region (SA) and African rice-producing region (AF) than in the other rice-producing regions, and the rates of increase were significantly higher than those in the other two rice-producing regions. The rate of increase under the high-emissions shared socioeconomic pathway (SSP585) was significantly higher than that under the low-emissions shared socioeconomic pathway. In the moderate-emissions shared socioeconomic pathway (SSP245), the number of warm days (Tx90p) in South American rice-production areas (SA) in 2050 increased by 58% compared with the historical reference period and increased by up to 163% in 2100 compared with the historical period.

The number of cold nights in the future (Tn10p) decreased significantly in the four rice-producing regions. Especially under the high-emissions shared socioeconomic pathway (SSP585) in South American rice-producing regions (SA) and African rice-producing regions (AF), cold night disasters similar to those in the historical period rarely occurred. In the moderate-emissions shared socioeconomic pathway (SSP245), the number of cold nights in the South American rice-producing region (SA) was 70% lower in 2050 and 136% lower in 2100 than that in the historical period. The number of cold nights in the African rice-producing region (AF) in 2050 and 2100 decreased by 58% and 127% from the historical period, respectively.

The total amount of heavy rainfall in the future (R99p) and the maximum rainfall on the fifth day (Rx5day) in South America (SA) and Southeast Asia (SEA) were significantly higher than those of other rice-producing regions. They increased significantly under the moderate-emissions shared socioeconomic pathway (SSP245) and the high-emissions shared socioeconomic pathway (SSP585). In the middle-emissions shared socioeconomic pathway (SSP245), the rainfall in the South American rice-producing region (SA) increased by 25% and 41% in 2050 and 2100, respectively, compared to the historical period. The rainfall in the Southeast Asian rice-producing region (SEA) increased by 23% and 38% in 2050 and 2100, respectively, compared with the historical period.

### 3.2. Relationship between Historical Rice Yield and Various Factors

Figure 5 mainly shows the relationship between the historical rice yield and extreme climatic events and economic factors in the Southeast Asian rice-producing region (SEA). In this region, rice yield had a good linear relationship with the GDP, land use, and population. There was no significant linear relationship between the effects of climatic factors on the rice yield.

Appendix A show the relationship between historical rice yields and extreme climatic events and economic factors in the African rice-producing region (AF), North American rice-producing region (NA), and South American rice-producing region (SA), respectively. The results of these analyses were similar to those for the Southeast Asian rice-producing region (SEA). The development of economic factors effectively promoted an increase in rice yield, and there was a good linear relationship. The influence of the extreme climate index was relatively weak, and its influence was nonlinear. In addition, the effects of extreme climatic events on the rice yield in different rice-producing regions were quite different. Based on this, we established different statistical yield models for the four production areas to explore future changes in the rice yield in different rice-production areas and the contribution of extreme climate events to the rice yield.

### 3.3. Future Rice Yield Changes under Different Temperature Rise Scenarios

In the previous section, we explored the relationship between historical extreme weather events, economic data, and rice yields. We constructed nonlinear models for four major rice-producing regions to explore the effects of climate change and economic changes on regional rice yields. Through the analysis of the model results, we found that the model can better simulate the historical output and apply the model to predict the future rice yield.

Table 1 shows that the future yield distribution of the four major rice-producing regions was similar to that in the historical period, and the yield per unit area of the rice-producing regions in North America and Southeast Asia was still significantly higher than that in the other two production areas. The main reason for this was that the hydrothermal conditions in the rice-producing regions of Southeast Asia were suitable for rice growth, while the developed economic base of the rice-producing regions in North America had a strong role in promoting rice yield. In the mid-21st century, the rice yields of the four major rice-producing regions were all projected to show different degrees of increase compared with the historical period. This was because economic development can still effectively and significantly increase rice production in the next few decades.

In the late 21st century, compared with the middle of the 21st century, the rice-production areas of Southeast Asia and South America were projected to have lower yields. However, the yields in the African rice-producing regions and the North American rice-producing regions increased, especially in the North American production areas. This was mainly because, in the late 21st century, the promotion effects of economic development and technological progress on the rice yield slowed down, and the impact of extreme climatic events on the yield increased. Combined with the rice-planting areas, Southeast Asia’s rice-producing regions will still be the most important rice-producing regions in the world in the future. The statistical yield model of the four regions also showed that economic factors (population, GDP, rice planting area) all had significantly positive effects on the future rice yield. An increase in extreme climate events was predicted to inhibit an increase in rice production. However, compared with economic factors, the influence of climatic factors was weaker. The increase or decrease in the number of warm days (Tx90p), the total amount of extremely heavy precipitation (R99p), and the maximum precipitation on the fifth day (Rx5day) reduced the rice yield to a certain extent. Compared with economic factors, however, these negative effects were not significant.

## 4. Discussion

The improvement of agronomic and economic management can significantly increase global rice production in existing planting areas [4]. Due to the lack of future forecast data, the economic factors selected for this study mainly include the GDP, land use, and population and do not fully describe the future agricultural economic system. The reliability of future economic factors is influenced by politics, policies, and other factors, and there is great uncertainty about its reliability. Rice yield cannot be increased indefinitely. There is a certain threshold for rice yield and the positive effect of economic factors, but this limitation is not considered in this paper.

## 5. Conclusions

Based on historical economic data and climate data, this paper uses a statistical yield model to construct a model of the impact of extreme climate events and economic factors on rice yield and analyzes the impact of extreme events on the rice yield in four major rice-producing regions of the world. Using the climate data output by the future CMIP6 model, future changes in extreme climate events in the four major rice-producing regions of the world are predicted. Based on historical models, future rice yields in these regions are estimated. The following conclusions are drawn:

(1) In the four rice-producing regions under different shared socioeconomic paths, the number of warm days (Tx90p), total heavy rainfall (R99p), and 5-day maximum rainfall (Rx5day) show an increasing trend in the future. The number of cold nights (Tn10p) shows a decreasing trend in the future. Extreme heat events increase significantly in all regions, and extremely low-temperature events are reversed. The extreme precipitation indices in the South American rice-producing regions (SA) and Southeast Asia rice-producing regions (SEA) are significantly higher than those in the other rice-producing regions. In the moderate-emissions shared socioeconomic pathway (SSP245), the South American rice-producing region (SA) increases by 25% and 41% in 2050 and 2100, respectively, compared to the historical period. The Southeast Asian rice-producing region (SEA) will increase by 23% and 38% in 2050 and 2100, respectively, compared with the historical period.

(2) In the four major rice-producing regions, historical rice yield has a good linear relationship with the GDP, land use, and population. There was no significant linear relationship between the effects of climatic factors on the rice yield. However, the increase in extreme climate events has a significant negative effect on rice yield [33,34].

(3) In the mid-21st century, the rice yields in the four major rice-producing regions all show an increasing trend to varying degrees compared with the historical period. At the end of the 21st century, the production of rice in Southeast Asia and South America decreased. However, the yields in the African and North American rice-producing regions increase, especially in the North American production areas. This is mainly because, in the late 21st century, the positive effect of economic development and technological progress on rice yield slows down, and the impact of extreme climatic events on the yield increases. The statistical yield model of the four regions also shows that economic factors (population, GDP, rice planting area) all have a significant positive effect on the future rice yield [4]. An increase in extreme climate events inhibits an increase in rice yield. The increase or decrease in the number of warm days (Tx90p), the total amount of extremely heavy precipitation (R99p), and maximum precipitation on the fifth day (Rx5day) reduce the rice yield to a certain extent, but the negative effects of economic factors are not significant. Therefore, government agencies should not only focus on the development of science and technology but also pay attention to the issue of climate change. Specifically, in the rice-producing areas of the Southeast Asia rice-producing region, it is necessary to increase the ability to prevent extreme high-temperature events and extreme precipitation in the future [16].

## Figures and Tables

**Figure 1 ijerph-19-04437-f001:**
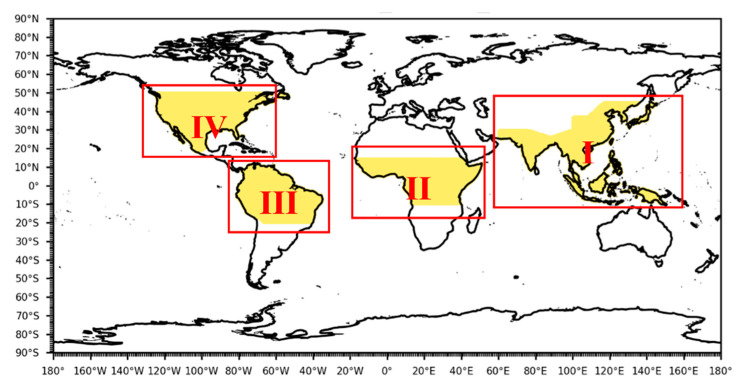
Division of the four major rice-producing regions in the world.

**Figure 2 ijerph-19-04437-f002:**
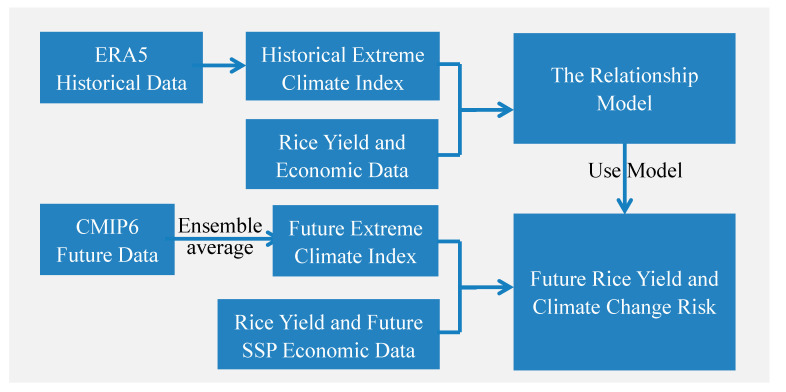
Flow chart of the impact of extreme climate events on future rice yields.

**Figure 3 ijerph-19-04437-f003:**
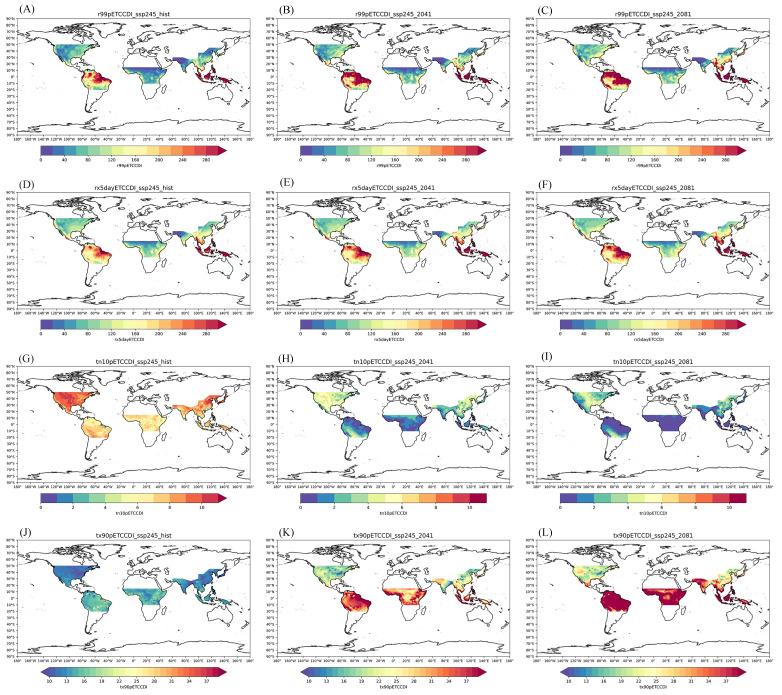
Spatial distribution of extreme climate events (R99p, Rx5day, Tn10p, Tx90p) in the world’s four major rice-producing regions under the SSP245 pathway. ((**A**–**C**) represent the historical R99p, the R99p of mid-21st century and the R99p of end of the 21st century, separately. Similarly, (**D**–**F**) represent Rx5day. (**G**–**I**) represent Tn10p. (**J**–**L**) represent Tx90p).

**Figure 4 ijerph-19-04437-f004:**
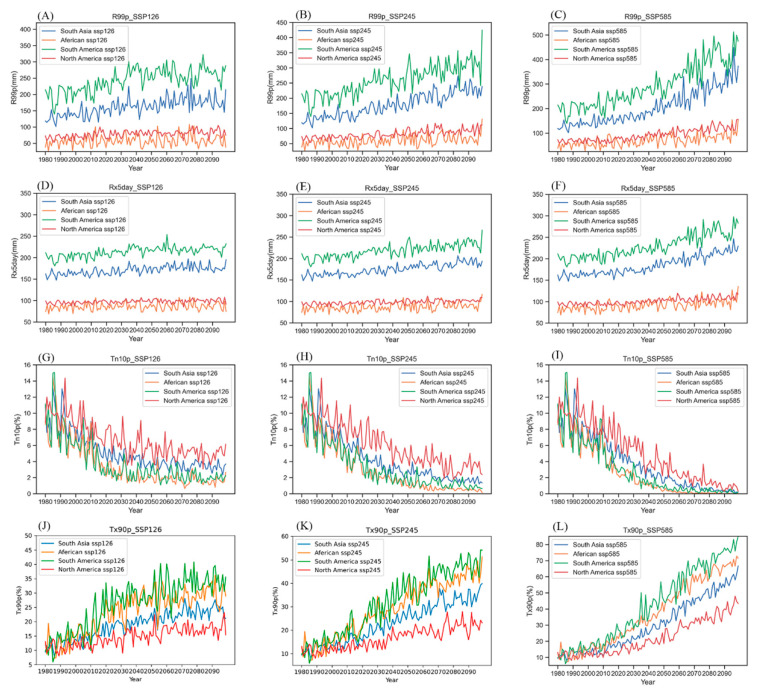
Historical and future changes in the regional averages of four extreme climate events (R99p, Rx5day, Tn10p, Tx90p) in the four rice-producing regions under different scenarios. ((**A**–**C**) represent the historical R99p, the R99p of mid-21st century, and the R99p of end of the 21st century, separately. Similarly, (**D**–**F**) represent Rx5day. (**G**–**I**) represent Tn10p. (**J**–**L**) represent Tx90p).

**Figure 5 ijerph-19-04437-f005:**
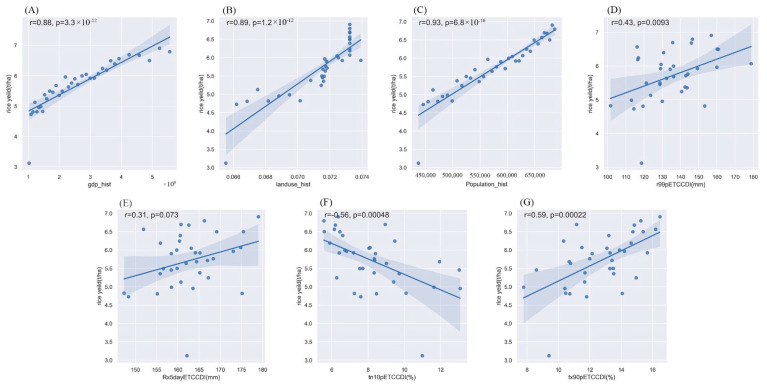
The relationship between extreme climate indices, economic factors, and rice yield in the Southeast Asia rice-producing region. (**A**) is the relationship between rice yield and GDP, (**B**) is the relationship between rice yield and land use, (**C**) is the relationship between rice yield and population, (**D**) is the relationship between rice yield and R99p, (**E**) is the relationship between rice yield and Rx5day, (**F**) is the relationship between rice yield and Tn10p, and (**G**) is the relationship between rice yield and Tx90p.

**Table 1 ijerph-19-04437-t001:** The historical rice yield and the prediction of future rice yield in four major rice-producing regions.

Rice-Producing Regions	Historical Yield (t/ha)	Yield of SSP126 (2050) (t/ha)	Yield of SSP245 (2050) (t/ha)	Yield of SSP585 (2050) (t/ha)	Yield of SSP126 (2100) (t/ha)	Yield of SSP245 (2100) (t/ha)	Yield of SSP585 (2100) (t/ha)
Southeast Asian rice-producing region (SEA)	5.71	5.83	5.82	5.78	5.63	5.57	5.58
African rice-producing region (AF)	1.85	1.94	1.98	1.97	2.00	1.97	1.97
South American rice-producing region (SA)	4.13	4.42	4.40	4.37	4.00	4.02	3.96
North American rice-producing region (NA)	6.94	6.89	6.88	6.88	8.22	8.21	8.21

## Data Availability

The CMIP6 data and ERA5 reanalysis are downloaded online from https://esgf-node.llnl.gov/projects/cmip6/ (accessed on 21 May 2021) and https://cds.climate.copernicus.eu/cdsapp#!/dataset/reanalysis-era5-single-levels?tab=overview (accessed on 11 April 2021). The Population, GDP, and Land-Use Data are downloaded online from https://esg.pik-potsdam.de/search/isimip/ (accessed on 15 May 2021).

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
