# Peer review of "Impacts of Extreme Climate Events on Future Rice Yields in Global Major Rice-Producing Regions"

_ijerph, 2022, doi:10.3390/ijerph19084437_

Round 1

Reviewer 1 Report

  • Overall, the MS need be Improved and Rewritten
  • The MS needs to be proofed by professional proofers.
  • See suggestion attached file:
  • Refine the title, Abstract and methods
  • The Figures quality are presented poorly (Pay attention on the size of letter and notation)
  • Insert a short and clear introduction before presenting the Table 2 or Figure (Figure 3, 4 and 5). Refine the title of Table 1 (The title should clear and concisely)
  •  Rewrite discussion seperately from conclusion
  • Summarize Discussion (without discussion)

Author Response

Manuscript ID: ijerph-1618447

Title: Impacts of Extreme Climate Events on Future Rice Yields

Journal: International Journal of Environmental Research and Public Health, Special Issue "Climate Change, Health, and Equity"

Dear editors and reviewers,

Thank you for your letter and for the reviewers’ comments concerning our manuscript entitled “Impacts of Extreme Climate Events on Future Rice Yields” (ID: 655128). Those comments are all valuable and very helpful for revising and improving our paper, as well as the important guiding significance to our researches. We have studied the valuable comments from editors and reviewers carefully, and tried our best to make extensive revisions which marked in red in the paper. The point to point responds to the reviewers’ comments are listed as following:

Response to the reviewers’ comments:

Reviewer #1:

  1. Overall, the MS need be Improved and Rewritten. The MS needs to be proofed by professional proofers.
  • Response: We thank the reviewer for careful reading of our paper and for the helpful comment. We are fully aware that our English expression needs to be improved. Following the reviewer's suggestion, our manuscript has undergone English language editing by AJE. The text has been checked for correct use of grammar and common technical terms, and edited to a level suitable for reporting research in a scholarly journal.

  1. Refine the title, Abstract and methods.
  • Response: We thank the reviewer for careful reading of our paper and for the helpful comment. Following the reviewer's suggestion, we refine the title, Abstract and methods.

  1. The Figures quality are presented poorly (Pay attention on the size of letter and notation)
  • Response: We thank the reviewer for careful reading of our paper and for the helpful comment. Following the reviewer’s comment, we redrawn all the pictures in the manuscript. The revisions are in the revised manuscript. In addition we submit the original version of all picture.

  1. Insert a short and clear introduction before presenting the Table 2 or Figure (Figure 3, 4 and 5). Refine the title of Table 1 (The title should clear and concisely)
  • Response: We thank the reviewer for careful reading of our paper and for the helpful comment. This is also the issue that this paper needs to improve. Following the reviewer’s comment, we insert a short and clear introduction before presenting the Table 1. We add the introduction of Figure (Figure 3, 4 and 5) in the title of Figure.

  1. Rewrite discussion seperately from conclusion. Summarize Discussion (without discussion).
  • Response: We thank the reviewer for careful reading of our paper and for the helpful comment. Following the reviewer's suggestion, we divide the conclusion and discussion into two parts. In addition, we also added some discussions. The revisions are on line 388-404 in the revised manuscript.

Special thanks to reviewer for your good comments.

We have tried our best to improve the manuscript and made some changes in the new revised manuscript. Our manuscript has been polished by a native English speaker.

We appreciate for Editors and Reviewer’s warm work earnestly, and hope that the correction will meet with approval.

Once again, thank you very much for your comments and suggestions.

Sincerely,

Jieming Chou

Weixing Zhao

Reviewer 2 Report

Here are some comments on the article: 

The general approach of the article is correct as well as the expression of the models. It strikes me that after various modelling, the results are scarce. 

I am not saying that they are not valid, after all it is the conclusion and the result of your model, but it usually happens with this type of methodology that after a long period of time consumed in modeling, the results are scarce or inconclusive. 

It is OK for publishing. 

Author Response

Manuscript ID: ijerph-1618447

Title: Impacts of Extreme Climate Events on Future Rice Yields

Journal: International Journal of Environmental Research and Public Health, Special Issue "Climate Change, Health, and Equity"

Dear editors and reviewers,

Thank you for your letter and for the reviewers’ comments concerning our manuscript entitled “Impacts of Extreme Climate Events on Future Rice Yields” (ID: 655128). Those comments are all valuable and very helpful for revising and improving our paper, as well as the important guiding significance to our researches. We have studied the valuable comments from editors and reviewers carefully, and tried our best to make extensive revisions which marked in red in the paper. The point to point responds to the reviewers’ comments are listed as following:

Response to the reviewers’ comments:

Reviewer #2:

  1. It strikes me that after various modelling, the results are scarce. I am not saying that they are not valid, after all it is the conclusion and the result of your model, but it usually happens with this type of methodology that after a long period of time consumed in modeling, the results are scarce or inconclusive.
  • Response: We thank the reviewer for careful reading of our paper and for the helpful comment. This is also the issue that this paper needs to improve. Following the reviewer's suggestion, we add the detail of the method in the part Method. In addition, we also show the model which is develop by historical data. The revisions are on line 313-318 in the revised manuscript.

Special thanks to reviewer for your good comments.

We have tried our best to improve the manuscript and made some changes in the new revised manuscript. Our manuscript has been polished by a native English speaker.

We appreciate for Editors and Reviewer’s warm work earnestly, and hope that the correction will meet with approval.

Once again, thank you very much for your comments and suggestions.

Sincerely,

Jieming Chou

Weixing Zhao

Reviewer 3 Report

The topic of the research is relevant from the aspect of global food security. The aim of the paper is well defined. The method and research design are appropriate. The analysis carried out is consistent. Results are presented clearly. But I strongly recommend checking the paper with a native proofreader. The English of the paper is a bit poor, using a lot of word repetition (for example, the verb 'show' appears four times in the abstract).
Keywords can be extended by words identifying the geographical characteristics of the paper.
Policy recommendations are needed in the section of 'Conclusion and discussion'.

Author Response

Manuscript ID: ijerph-1618447

Title: Impacts of Extreme Climate Events on Future Rice Yields

Journal: International Journal of Environmental Research and Public Health, Special Issue "Climate Change, Health, and Equity"

Dear editors and reviewers,

Thank you for your letter and for the reviewers’ comments concerning our manuscript entitled “Impacts of Extreme Climate Events on Future Rice Yields” Those comments are all valuable and very helpful for revising and improving our paper, as well as the important guiding significance to our researches. We have studied the valuable comments from editors and reviewers carefully, and tried our best to make extensive revisions which marked in red in the paper. The point to point responds to the reviewers’ comments are listed as following:

Response to the reviewers’ comments:

Reviewer #3:

  1. But I strongly recommend checking the paper with a native proofreader. The English of the paper is a bit poor, using a lot of word repetition (for example, the verb 'show' appears four times in the abstract).
  • Response: We thank the reviewer for careful reading of our paper and for the helpful comment. We are fully aware that our English expression needs to be improved. Following the reviewer's suggestion, our manuscript has undergone English language editing by AJE. In addition, we rewrite the abstract. The text has been checked for correct use of grammar and common technical terms, and edited to a level suitable for reporting research in a scholarly journal.

  1. Keywords can be extended by words identifying the geographical characteristics of the paper.
  • Response: We thank the reviewer for careful reading of our paper and for the helpful comment. Following the reviewer's suggestion, we add a new keyword of the paper.

  1. Policy recommendations are needed in the section of 'Conclusion and discussion'.
  • Response: We thank the reviewer for careful reading of our paper and for the helpful comment. Following the reviewer's suggestion, we divide the conclusion and discussion into two parts. In addition, we also added some Policy recommendations in discussion. The revisions are on line 388-404 in the revised manuscript.

Special thanks to reviewer for your good comments.

We have tried our best to improve the manuscript and made some changes in the new revised manuscript. Our manuscript has been polished by a native English speaker.

We appreciate for Editors and Reviewer’s warm work earnestly, and hope that the correction will meet with approval.

Once again, thank you very much for your comments and suggestions.

Sincerely,

Jieming Chou

Weixing Zhao

Round 2

Reviewer 1 Report

In general, this MS need to improved and the authors should follow the guidance (instruction for the authors)

FOLLOW THE GUIDLINE (INSTRUCTION TO AUTHORS)

  • INTRODUCTION
  • MATERIALS AND METHODS
  • RESULTS
  • DISCUSSION
  • CONCLUSION

Citation and Reference: Please follow the guidance strictly and adopted in whole MS

In the text, reference numbers should be placed in square brackets [ ], and placed before the punctuation; for example [1], [1–3] or [1,3]. For embedded citations in the text with pagination, use both parentheses and brackets to indicate the reference number and page numbers; for example [5] (p. 10). or [6] (pp. 101–105).

FIGURE 1-5:  Change to Figure (Not all the word Capitalized)

All Figures, Schemes and Tables should be inserted into the main text close to their first citation and must be numbered following their number of appearance (Figure 1, Scheme I, Figure 2, Scheme II, Table 1, etc

Line: 208: RESULT AND DISCUSSION Change to RESULTS:  In this section you only presenting the RESULTS no Discussion (see guidance)

Line: 395 DISCUSSIONS (Presented after RESULTS

Highlight of your results and finding and compare or support with the reference. In this paper,  no reference is mentioned..  The findings and their implications should be discussed in the broadest context possible and limitations of the work highlighted. Future research directions may also be mentioned.

CONCLUSSION (Conclusion should be presented before acknowledgments) See the guidance

Please summarized the discussion to 200 – 300 words, presenting only the highlight of your finding  

Author Response

Manuscript ID: ijerph-1618447

Title: Impacts of Extreme Climate Events on Future Rice Yields in Global Major Rice-producing Regions

Journal: International Journal of Environmental Research and Public Health, Special Issue "Climate Change, Health, and Equity"

Dear editors and reviewers,

Thank you for your letter and for the reviewers’ comments concerning our manuscript. Those comments are all valuable and very helpful for revising and improving our paper, as well as the important guiding significance to our researches. We have studied the valuable comments from editors and reviewers carefully, and tried our best to make extensive revisions which marked in red in the paper. The point to point responds to the reviewers’ comments are listed as following:

Response to the reviewers’ comments:

Reviewer #1:

  1. FOLLOW THE GUIDLINE (INSTRUCTION TO AUTHORS)
  • Response: We thank the reviewer for careful reading of our paper and for the helpful comment. Following the reviewer's suggestion, we reread the “INSTRUCTION TO AUTHORS” and our manuscript has been revised again.

  1. In the text, reference numbers should be placed in square brackets [ ], and placed before the punctuation; for example [1], [1–3] or [1,3]. For embedded citations in the text with pagination, use both parentheses and brackets to indicate the reference number and page numbers; for example [5] (p. 10). or [6] (pp. 101–105).
  • Response: We thank the reviewer for careful reading of our paper and for the helpful comment. Following the reviewer's suggestion, we change the style of reference in the revised manuscript.

  1. FIGURE 1-5: Change to Figure (Not all the word Capitalized)

  • Response: We thank the reviewer for careful reading of our paper and for the helpful comment. Following the reviewer's suggestion, we change the name of all figures in the revised manuscript.

  1. All Figures, Schemes and Tables should be inserted into the main text close to their first citation and must be numbered following their number of appearance (Figure 1, Scheme I, Figure 2, Scheme II, Table 1, etc
  • Response: We thank the reviewer for careful reading of our paper and for the helpful comment. This is also the issue that this paper needs to improve. Following the reviewer's suggestion, all Figures, Schemes and Tables inserted into the main text close to their first citation.

  1. Line: 208: RESULT AND DISCUSSION Change to RESULTS: In this section you only presenting the RESULTS no Discussion (see guidance)

Line: 395 DISCUSSIONS (Presented after RESULTS)

  • Response: We thank the reviewer for careful reading of our paper and for the helpful comment. Following the reviewer's suggestion, we change “RESULT AND DISCUSSION Change” to “Results”, and change the location of “Conclusion” and “Discussion”

  1. Highlight of your results and finding and compare or support with the reference. In this paper, no reference is mentioned..  The findings and their implications should be discussed in the broadest context possible and limitations of the work highlighted. Future research directions may also be mentioned.
  • Response: We thank the reviewer for careful reading of our paper and for the helpful comment. Following the reviewer's suggestion, we highlight our results and finding and compare with the reference.

  1. CONCLUSSION (Conclusion should be presented before acknowledgments) See the guidance
  • Response: We thank the reviewer for careful reading of our paper and for the helpful comment. Following the reviewer's suggestion, we read the guidance and change in the revised manuscript.

  1. Please summarized the discussion to 200 – 300 words, presenting only the highlight of your finding
  • Response: We thank the reviewer for careful reading of our paper and for the helpful comment. Following the reviewer's suggestion, we summarized the discussion.

Special thanks to reviewer for your good comments.

We have tried our best to improve the manuscript and made some changes in the new revised manuscript. Our manuscript has been polished by a native English speaker.

We appreciate for Editors and Reviewer’s warm work earnestly, and hope that the correction will meet with approval.

Once again, thank you very much for your comments and suggestions.

Sincerely,

Jieming Chou

Weixing Zhao
